# Effects of Combining High Pressure Processing Treatments and Konjac Glucomannan and Sodium Caseinate on Gel Properties of Myosin Protein

**DOI:** 10.3390/foods12040691

**Published:** 2023-02-05

**Authors:** Yingying Cao, Lila Zhao, Huaiyu Li

**Affiliations:** College of Life Science and Engineering, Lanzhou University of Technology, Lanzhou 730050, China

**Keywords:** high pressure, myosin protein, konjac glucomannan, sodium caseinate, scanning electron microscopy

## Abstract

Effects of two high pressure processing treatments and various levels of konjac glucomannan (KGM) and sodium caseinate (SC) on texture properties, water-holding capacity, and ultra-structure of gels of rabbit myosin protein were investigated. The two high pressure processing treatments were as follows: (1) mean pressure (200 MPa), low temperature (37 °C), and holding for a short time (5 min) followed heating (80 °C for 40 min) (gel LP + H), and (2) high pressure (500 MPa), high temperature (60 °C), and holding for a long time (30 min) (gel HP). Gel LP + H have better gel properties (increased hardness, springiness, gumminess, adhesiveness, cohesiveness, and water binding capacity) than gels HP. Above all, gels myosin + SC:KGM (2:1) have best gel properties. KGM and SC both significantly improved the gel texture properties and water binding capacity.

## 1. Introduction

Gelation is an important functional property of meat protein, which affects the quality of the product. Myosin is responsible for a number of properties such as gelation, water holding, and emulsification. Gel formation involves partial denaturation of protein followed by irreversible aggregation, which results in a three dimensional network [1]. 

High pressure (HP) has been showed to produce foods of natural-like characteristics or possessing modified food characteristics [2]. There has been great interest in the application of HP to selectively denature proteins, which might otherwise lead to a decrease in the solubility of proteins and changes in functional properties; for example, gel-forming ability and water holding capacity (WHC) [3]. Moreover, HP-induced gels that affect food texture are more deformable and softer than heat-induced gels.

Sodium caseinate (SC) is a widely used food ingredient. Kuraishi et al. [4] found that SC appeared to be a superior substrate for the cross-linking to meat proteins than soy protein, whey protein, or gelatin. It has been showed to be an excellent substrate of SC for inducing cross-linking in restructured meat products [5]. 

Konjac glucomannan (KGM) is a high-molecular weight, water soluble, non-ionic polysaccharide extracted from Amorphophallus konjac, a plant native to East Asia, where it has been used since ancient times. KGM is a neutral plant polysaccharide used as a stabilizer, thickener, or gelling agent in foods. Multidimensional statistical analysis shows that the konjac research developed rapidly in food [6]. KGM is comprised of blocks of β-1, 4-linking D-mannose and D-glucose at a ratio of 1.6:1 (number of D-mannose/number of D-glucose), with approximately 1/19 of the sugar units being acetylated [7,8]. It has been generally recognized as safe by a consensus of scientific opinions since 1994 [9,10]. KGM has a dual functionality; on the one hand, it is highly beneficial for human health [11]. KGM cannot be hydrolyzed by digestive enzymes in the human upper gastrointestinal tract and, therefore, is considered as a non-calorie indigestible dietary fiber. It has been showed to play a role in weight reduction [12,13]. Its function was reported to prevent and treat constipation [14], regulate lipid metabolism, improve glucose metabolism, and reduce the risk of developing diabetes and heart disease [15]. On the other hand, KGM can be used as various kinds of food additives because of its good water absorptive capability [16], gel-forming ability [17], stability, emulsifying ability, thickening ability, and forming film properties [18,19]. KGM gel can be used to simulate fat characteristics and reduce the fat content of meat products, can be used to reinforce the gels from low-quality squid surimi, and to improve textural and water holding properties [20]. KGM itself has a strong water-binding ability; when combined with other hydrocolloids, it exhibits a synergistic effect on protein gelation and water-holding in comminuted meat products [21]. In addition, acetylated (dds) konjac glucan an (KGM) can make the microstructure of the gel more compact and more uniform [22].

This study was based on evaluating the high pressure-heating of myosin protein gels in response to adding KGM and SC by investigating the changes of texture, water binding capacity, and ultra-microstructure to research the interaction between protein and water-soluble colloids and to promote the application of water-soluble colloids in the meat industry. The other aim was to research the combined effects of high pressure and heating treatments to myosin gel characteristics.

## 2. Materials and Methods

KGM was supplied by Nong-Tai Company, Hubei province, China. SC was supplied by Shidai-Tianjiao Company, Zhengzhou, Henan province, China. Both were dry powder. Rabbits were supplied by the rabbit nursery of Jiangsu Academy of Agricultural Sciences. Myosin was extracted from rabbits (6 months old, weighing approximately 3.5 kg) according to the methods described by Wang et al. and Hermansson et al. [23,24] with some modifications. The whole extraction process was completed at 4 degrees below zero. The peeled psoas major muscle was weighed as 100 g, 500 mL of Rigor buffer (0.1 mol·L^−1^ KCl, 20 mmol·L^−1^ potassium phosphate, 2 mmol·L^−1^ EGTA, pH 7.0) and DTT (1 mmol·L^−1^) were added, followed by grinding with a high-speed tissue crusher. Triton X-100 (1%) was added, followed by gentle stirring for 10 min and centrifugation at 8000× *g* for 10 min. The precipitate was collected, 300 mL of Rigor buffer was added, and was gently stirred for 10 min followed by centrifugatation at 8000× *g* for 10 min. The precipitate was collected and 300 mL of Tris-Buffer (5 mmol·L^−1^ Tris, 1 mmol·L^−1^ EGTA, pH 7.8) was assed to disperse the precipitate, so that the final concentration of DTT was 1 mmol·L^−1^.

### 2.1. Sample Preparation

The myosin solution was at the concentration of 20 mg/mL, 0.6 M NaCl, pH 6.75, and contained 20 mM sodium phosphate. Various concentrations of SC and KGM were slowly added to the myosin proteins (Table 1). The myosin:SC in the gelling solute was 15:4, which mimics a typical emulsified-sausage [25]. 

### 2.2. High-Pressure Processing

High-pressure processing was carried out in a high-pressure unit (HPP 600 MPa/4 L, Baotou Kefa, Baotou, China) equipped with a temperature and pressure regulation. Water was used as the pressure-transfer medium. Prior to pressure processing, the mixed solution was filled in 2 mL centrifuge tubes and sealed by a cover without air bubbles. The high pressure was increased at speed of 3.5 MPa/s to required values, and released within 5 s. The following treatments were applied: (a) (gel LP + H) mean pressure (200 MPa), lower temperature (37 °C), and holding for a short time (5 min) (non-denaturing temperature, as the pressurizing medium) followed (after pressure release) by heating of the sample in a water bath at 80 °C for 40 min, and (b) (gel HP) heating under pressure conditions: high pressure (500 MPa), higher temperature (60 °C), and holding for a long time (30 min) (pressurization at denaturing temperature). After heating and high pressure treating, the samples were heated to 70 °C for 20 min and then immediately chilled with cool water and stored in a 4 °C room for 48 h before analysis. 

### 2.3. Texture Profile Analysis (TPA)

Before measurement, the samples were placed at room temperature for 2 h. The texture was determined using the Texture Profile Analysis method (five replicates treatment) with Texture Analyzer (TA-XT plus, Stable Micro Systems, Surrey, UK). Samples were compressed twice with a P5 probe (5 mm cylinder Stainless) at a pre-test Speed of 1.0 mm/s, test speed of 1 mm/s, post-test speed of 2 mm/s, trigger force of 5 g, and a deformation distance of 15 mm. The samples were placed in 10 mL centrifuge tubes with a height of 8 cm.

Attributes were calculated as follows: hardness (N), peak force required for first compression; springiness (mm), distance the sample recovers after the first compression; adhesiveness (g), the negative force area for the first bite representing the work necessary to pull the compressing plunger away from the sample; cohesiveness (dimensionless), ratio of positive force area during the second compression; gumminess (g), the product of hardness and cohesiveness; resilience (dimensionless), the capacity of the gel return to its original shape or position after deformation.

All the results were analysis by the software of Texture Expert Exceed 2.64a TPA.MAC. Measurements were carried out at room temperature.

### 2.4. Water Binding Capacity (WBC) 

The WBC of gels has an impact on several quality traits and represents the ability of proteins to bind to and retain water. This property is also largely used to evaluate the quality and yield of meat products. 

A frozen sample (about 2 g) w_s_ was placed in a 7 mL centrifuge tube w_t_, then centrifuged in a Beckman Avanti J-E centrifuge (Beckman Coulter, Fullerton, CA, USA). All samples were centrifuged for 10 min at 10,000× *g* at 4 °C. After centrifuging, the water was drawn off with filter paper. Then gels were weighed w_g_. WBC was expressed as percent water retained per 100 g water present in the sample prior to centrifuging. Treatments were performed in triplicate and measurements were carried out in triplicate, in order to understand the stability of gels. WBC was measured by a modification of a published method [26]. WBC = (w_g_ − w_t_/w_s_ − w_t_) × 100%.

### 2.5. Scanning Electron Microscopy (SEM)

The microstructure of myosin gels was examined by scanning electron microscopy (SEM). The samples were fixed in 0.1 M glutaraldehyde in phosphate buffer (pH 7.4) for 2 h and then dehydrated in an increasing series of ethanol:water ratios (50%, 70%, 90%, 95%, 100%). The specimens were finally transferred into tertiary butyl alcohol and freeze-dried and coated with gold to a thickness of 10 nm. Samples were observed at an accelerating voltage of 15 kV and a magnification of 2000 using a scanning electron microscope (S-3000N, Hitachi, Tokyo, Japan).

### 2.6. Statistical Analysis

The data were analyzed using a one-way analysis of variance (ANOVA), and in some determinations, using the Tukey test, with SPSS 13.0. The confidence level was set for *p* < 0.05 level.

## 3. Results and Discussion

### 3.1. Texture Profile Analysis (TPA)

Different numbers (1 or 2) in the same sample indicate significant differences (*p* < 0.05) between HP and LP + H treatment. Different letters (a, b, c or d) in the same batch indicated significant differences (*p* < 0.05) among the sample. The Figure 1, Figure 2, Figure 3, Figure 4 and Figure 5 were the same.

The hardness value of the different treatments was showed in Figure 1. Gel LP + H presented with an increased hardness than the HP sample (*p <* 0.05). These results seem to indicate that higher temperature induced more cross linkage between the myosin proteins and SC and KGM. The control (C) gel had lowest hardness, while C + SC + mKGM had the highest values. SC did not increase the hardness of gels while KGM significantly increased their hardness. Pietrasik and Jarmoluk [27] used SC and k-carrageenan (CGN) to research binding and textural properties of pork muscle gels; they reported that SC can increase the effect on hardness, which was dependent on the content of CGN in homogenates. This might mean that SC did not increase gel hardness but can combine with other colloids to increase the hardness of mixed gels. Hermansson [28] had the perspective that SC cannot form self-supporting gels but rather acts as a paste in meat systems. The reason proposed for this difference in hardness is that SC is a kind of non-gelling additive that is not capable of forming an undivided network structure; however, KGM is a gelling polysaccharide that is able to form a network structure. Furthermore, the hardness of gels that contained SC and KGM at a ratio of 2:1 was higher than those with a ratio of 1:1. These results indicated that different ratios of SC and KGM in myosin mixed gels resulted in a different interactive effect. We supposed that KGM, water, and myosin aggregated and formed a network structure by hydrogen bonding, molecular dipoles, transient dipoles, and so on.

The springiness properties of the gels are shown in Figure 2. The HP sample springiness values varied from 0.200 to 0.871 mm and the LP + H sample values varied from 0.484 to 0.921 mm. The control showed the lowest springiness values while the C + SC + mKGM sample showed the highest values. Both SC and KGM significantly increased the springiness of gels, which was increased more in SC than in KGM. This may attribute to the good emulsifying ability of SC. The springiness of C + SC + mKGM and C + SC + sKGM in LP + H treatment had no obvious difference. While in the HP treatment, with increasing the level of KGM, the springiness was significant increased. 

The cohesiveness properties of the gels are shown in Figure 3. The cohesiveness values of LP + H treatment were slightly higher than HP treatment. The control sample had the lowest values while the C + SC + mKGM sample showed the highest values. Increasing the amount of KGM showed no obvious increase. This indicated that SC and KGM had a synergetic effect in myosin gel systems. Both SC and KGM can significantly increase the cohesiveness of gels, which was increased more by KGM than SC. These results indicated that SC and KGM significantly increased the viscosity and thickness of myosin protein gels. The force of the molecular structure became stronger. 

The adhesiveness properties of the gels are shown in Table 2. The adhesiveness of the control, C + SC, and C + SC + sKGM (HP treatment) was zero. In LP + H treatment, the adhesiveness of C + SC was low, while the adhesiveness of C + SC + mKGM was high. SC and KGM can significantly increase the adhesiveness of gels, especially in the C + SC + mKGM sample where the adhesiveness was rapidly increased by triple compared to the C + SC + sKGM treatment. A higher temperature did increase the adhesiveness between SC, KGM, and myosin proteins. These results indicated that SC and KGM increase the strength of the bonds between myosin and other ingredients.

The gumminess properties of the gels are shown in Figure 4. The gumminess of myosin gels was very low. When adding SC, the gumminess of the gel showed no significant increase; however, when adding small amounts of KGM, the gumminess of gels significantly increased in both HP and LP + H samples. This may suggest that a higher pressure (500 MPa) can improve the gumminess of KGM compared to the 200 MPa pressure. SC did not increase the gumminess of gels, while KGM can significantly increase their gumminess. KGM has a very high gumminess; it can stick to myosin proteins to form compact network gels.

The resilience properties of the gels are shown in Figure 5. The C + SC + sKGM gel showed the highest resilience. When adding SC, the resilience (0.075) significant increased (*p* < 0.01). When adding more KGM, the resilience significantly decrease. The resilience (0.64) in the control LP + H treatment was quite high, while the resilience of myosin was zero in the HP treatment. This suggested that only completely denatured myosin can display good resilience. SC significantly improved the resilience of the C + SC gel in HP treatment. Adding a small amount of KGM significantly increased the resilience, while adding more KGM significantly decreased the resilience. 

The results indicated that SC and a low level of KGM can improve gel resilience. Both additives had a synergistic reaction in the gel resilience. A high level of KGM decreased the gel resilience. The reason was that a high KGM content resulted in a very high viscosity of the gel, and when compressing, it has great pull force.

Comparing all the texture profile analyses, the values of HP samples were lower than the LP + H samples. It was suggested that 500 MPa, 60 °C, holding for 30 min were not enough to denature myosin, SC, and KGM; therefore, less interactions and cross-links were formed. It was indicated that a lower pressure (about 200 MPa) then binding heating treatment can obtain better TPA property gels in the meat industry.

### 3.2. Water Binding Capacity (WBC)

Water binding capacity of all gels is shown in Table 3. WBC has no significant differences at days 1 and 14, which indicated that combining konjac glucomannan, sodium caseinate, and high-pressure processing treatments was stable when stored at 4 °C.

In LP + H treatment, C gel exhibited the lowest water binding capacity at both 1 day and 14 days. There was significant increase in water binding capacity when adding SC. Atughonu et al. [29] reported that 2% sodium caseinate addition increased the cooking yield of frankfurters. Su et al. [30] reported that SC contributed to the formation of the protein network, thus enhancing the thermal stability of reduced-fat frankfurters. Pietrasik and Jarmoluk [27] found that the greater the SC content, the smaller the weight loss of pork muscle gels. Silva et al. [31] obtained same result in raw and cooked ham pâté where the addition of bovine blood globin that contained SC increased the WBC of all experiments.

The C + SC + sKGM gel showed the highest water binding capacity. It was demonstrated that SC and a small amount of KGM improved myosin gel WBC. It suggested that a protein-hydrocolloid interaction between the SC, KGM, and myosin had occurred. Herranz et al. [32], using alkalis on konjac glucomannan gels in restructured seafood products, indicated that KGM could reinforce the ability of the final product to capture moisture during cooking and retain its texture. Chin et al. [25] reported that when Konjac flour was added to the myofibrillar protein gel formulation, it completely prevented moisture loss from the cooking gel, even in those prepared at low salt concentrations. Chin et al. [25] found that higher levels of KGM could significantly (*p* < 0.05) enhance the water-holding properties of the surimi gels in comparison with the control (0% KGM). They thought it was likely due to the strong water absorptivity of KGM in the surimi gels. It was also observed that increasing KGM did not led to an increase in the WBC of the C + SC + mKGM systems. Lin and Huang (2008) reported that an increased konjac level significantly elevated the water-holding capacity for treatments of the same molecular-weight. Our results may be because there was not enough water and protein to bind, that was to say that KGM was an overdose. As a result, it probably led to less water affinity that was not trapped among myosin gel filaments and segments in C + SC + mKGM systems. 

In HP treatment, the WBC of C and C + SC could not be measured. This result is because the water could not be separated from the gel systems after centrifugation. We found that the C gel was in a solid state. In the C + SC sample, the gel was thicker than the control after centrifugation, and only very little water was separated. However, adding a small amount KGM resulted in a finer gel state (water could be separated from myosin, SC, and KGM after centrifugation). In the C + SC + mKGM gel, the mobile phase contained KGM and water. This result confirmed that in the C + SC + mKGM sample, KGM was as overdose.

Pietrasik and Jarmoluk [27] reported that synergistic interactions of carrageenan with myofibrillar proteins could exist through balanced electrostatic repulsion of like charges and attraction between polar or negatively charged groups and positively charged groups on proteins. They also indicated that the matrix formed in those gels had a greater ability to entrap water than that of other gels.

### 3.3. Scanning Electron Microscopy (SEM)

The physical attributes of heat-induced gel texture is highly dependent on the microstructure of the gel [33]. The evaluation of microstructural characteristics should provide valuable insight into the structural development of gel-forming proteins. The results of scanning electron microscopy showed differences in the three-dimensional microstructure of different treatments (Figure 6 and Figure 7).

It has been shown that 200 Mpa is the optimal pressure for hydrophobic interactions [34]. At 200 MPa, when only myosin in present, the gel was composed of disrupted, loose strands with an irregular order with many small cavities. When adding the SC, gels exhibited a fine gel matrix, consisting of a compact and smooth three-dimensional network with a uniform filament structure and higher cross linkage than only myosin. This seemed to indicate that a good SC-myosin interaction was formed. C + SC + sKGM gels exhibited a coarse structure with bigger cavities but a lesser network structure than C + SC + mKGM gels. Many small white balls “floated” in the protein matrix, which may be insoluble myosin. Whether an interaction exists between the myosin protein and the SC and KGB components cannot be determined from these micrographs. We only found that a good interaction exists between the myosin protein and SC and KGM.

At 500 MPa, only myosin resulted in a thread-like, weak, coarse, disorderly filamentous structure. When adding SC, gels displayed a compact, ordered, homogeneous three-dimensional network with many small cavities. This would indicate that protein–protein and protein-SC interactions, which strengthened the cross linkage of gels, were further established. SC resulted in the formation of better myosin gels. C + SC + sKGM gels exhibited many big, compact, orderly cavities. C + SC + mKGM gels had a macromolecular network that was less continuous, coarsely aggregated, resulting in a network with irregular and big cavities. The gels containing KGM exhibited a honeycomb feature, which was not seen in gels without KGM. Although the gels containing KGM showed a disordered, irregular, and tangled structure, but could bind more water (WBC were significant increase with adding KGM). Kuraishi et al. [4] also found the lack of consistent structural uniformity in KGM-treated myofibrillar gels, which seemed to reflect poor protein–polysaccharide interactions, which likely resulted from low protein solubility. The SEM results were supported by TPA and WBC.

All the SEM indicated that SC and KGM play an important role in forming the three-dimensional network structure through chemical bonding.

Comparing HP with LP + H treatments, the SEM showed the HP samples had less cross linkage, were less compact, less continuous, less disordered, less cavities, and were softer than LP + H samples. These microstructures support the TPA observation and WBC values.

## 4. Conclusions

Both KGM and SC significantly improved TPA properties (hardness, springiness, gumminess, adhesiveness, cohesiveness, and resilience) and the water-binding capacity in myosin mixed gels. The interaction between KGM and SC with myosin played an important role in the improved gelling properties of the composite gel systems. Gels created with a mean pressure (200 MPa), lower temperature (37 °C), and a short holding time (5 min) before heating to 80 °C for 40 min were better than gels at a high pressure (500 MPa), high temperature (60 °C), and holding for 30 min, regardless of the TPA properties or in water-binding capacity. In the meat industry, we can use the mean pressure (about 200 MPa) binding heating can gain a better product. Following comprehensive consideration, gels of SC and KGM (1:1) had the best gel properties. Overall, through the application of KGM and SC, it is possible to produce well-structured and better TPA property meat products with maximum water binding capacity.

## Figures and Tables

**Figure 1 foods-12-00691-f001:**
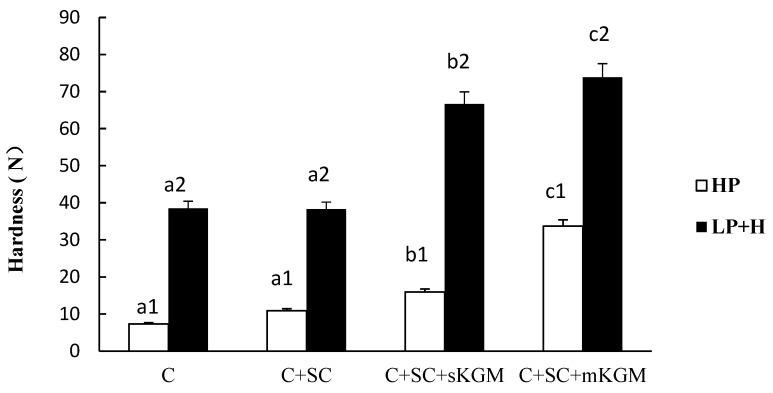
Effects of addition of various levels of SC and KGM on the hardness of different gels. Different numbers (1 or 2) in the same sample indicate significant differences (*p* < 0.05) among all treatments. Different letters (a, b, c or d) in the same batch indicated significant differences (*p* < 0.05) among the sample. The Figure 1, Figure 2, Figure 3, Figure 4 and Figure 5 were the same.

**Figure 2 foods-12-00691-f002:**
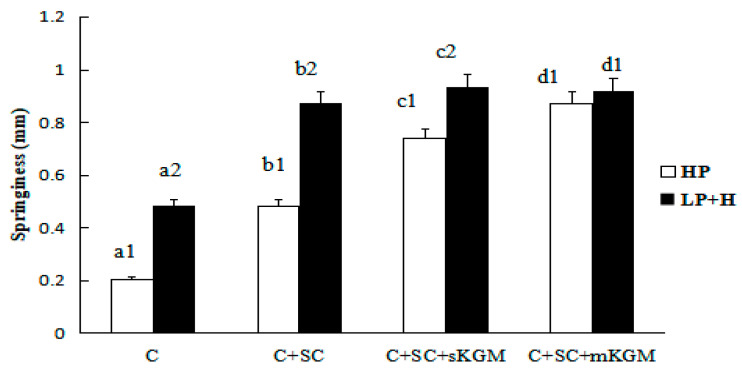
Effects of addition of various levels of SC and KGM on the springiness of different gels.

**Figure 3 foods-12-00691-f003:**
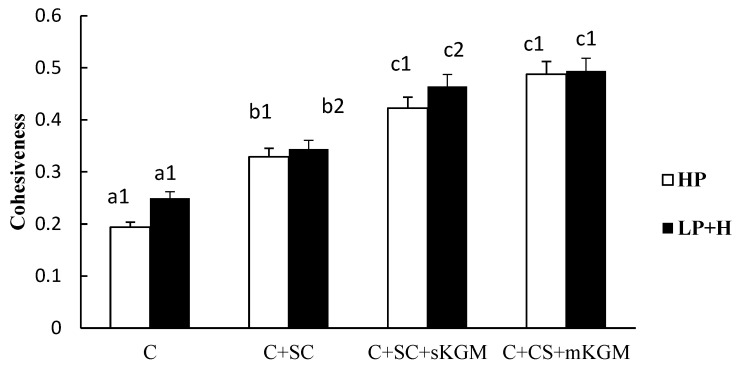
Effects of addition of various levels of SC and KGM on the cohesiveness of different gels.

**Figure 4 foods-12-00691-f004:**
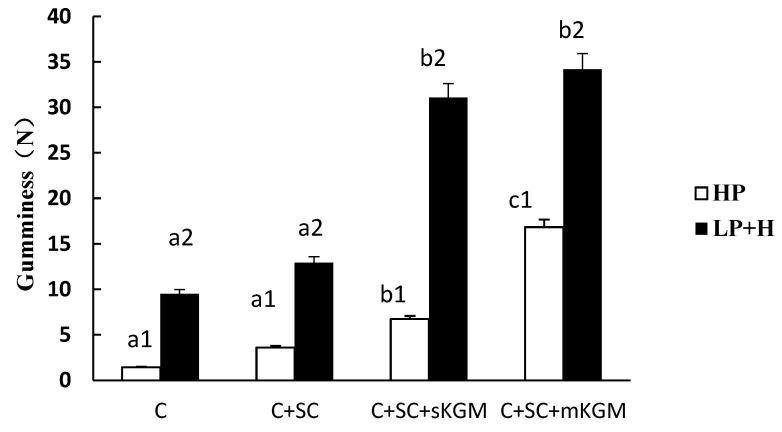
Effects of addition of various levels of SC and KGM on the gumminess of different gels.

**Figure 5 foods-12-00691-f005:**
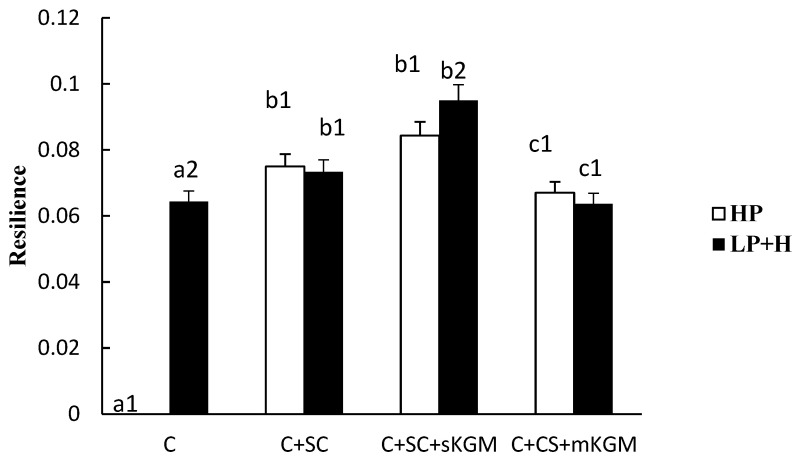
Effects of addition of various levels of SC and KGM on the resilience of different gels.

**Figure 6 foods-12-00691-f006:**
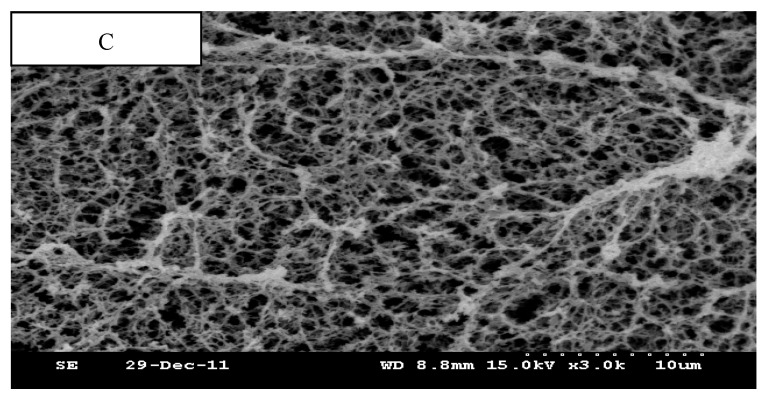
SEM micrographs (3000×) of 200 MPa LP + H rabbit myosin protein gels.

**Figure 7 foods-12-00691-f007:**
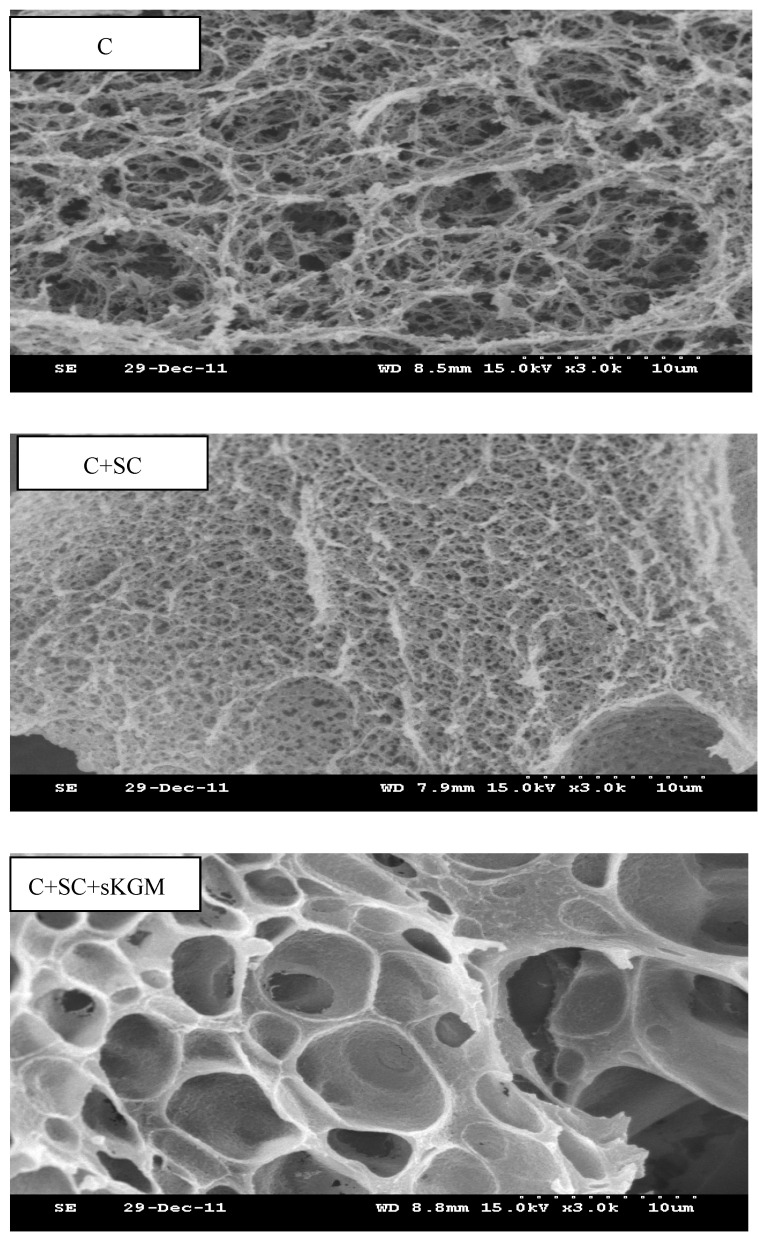
SEM micrographs (3000×) of 500 MPa HP + H rabbit myosin protein gels.

**Table 1 foods-12-00691-t001:** Experimental design showing the final concentrations of the specific ingredients in the gelling solutions.

Ingredient (mg/mL)	Myosin (mg/m)	Myosin + SC (mg/mL)	My + SC:KGM (1:1) (mg/mL)	My + SC:KGM (1:0.5) (mg/mL)
	(C)	(C + SC)	(C + SC + mKGM)	(C + SC + sKGM)
Myosin	20	20	20	20
SC	0	5.33	5.33	5.33
KGM	0	0	5.33	2.67

C: control; SC: sodium caseinate; KGM: konjac glucomannan; mKGM: many amount of KGM; sKGM: small amount of KGM. The gel was made three times.

**Table 2 foods-12-00691-t002:** Effects of addition of various levels of SC and KGM on the adhesiveness of different gels.

Adhesiveness (N.mm)	C	C + SC	C + SC + sKGM	C + SC + mKGM
HP	0	0	0	−15.79
LP + H	0	−9.53	−12.51	−45.55

**Table 3 foods-12-00691-t003:** WBC (water binding capacity) of myosin gels after 1 and 14 days of chilled storage.

	1 Day (%)	14 Days (%)
a (200 MPa):		
C	62.73 ^a1^ ± 2.94	60.47 ^a1^ ± 1.33
C + SC	77.73 ^b1^ ± 2.29	76.26 ^b1^ ± 0.79
C + SC + sKGM	86.93 ^c1^ ± 0.78	85.97 ^c1^ ± 0.86
C + SC + mKGM	87.67 ^c1^ ± 2.33	87.41 ^c1^ ± 1.20
b (500 MPa):		
C	---	---
C + SC	---	---
C + SC + sKGM	78.23 ^a1^ ± 0.23	77.97 ^a1^ ± 0.32
C + SC + mKGM	84.17 ^b1^ ± 0.78	83.30 ^b1^ ± 0.54

^a–c^ means in a same column with different letters are significant different (*p* < 0.05). Variation of the mean represents standard deviation of triplicate for each treatment. --- represents that the WBC could not be measured since it was not gel structure after centrifuging. Numbers (1) in the same sample indicate significant different (*p* < 0.05) between HP and LP + H treatment. Different letters (a, b) in the same batch indicated significant different (*p* < 0.05) among the sample.

## Data Availability

Data is contained within the article.

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
