# Peer review of "Effects of Combining High Pressure Processing Treatments and Konjac Glucomannan and Sodium Caseinate on Gel Properties of Myosin Protein"

_foods, 2023, doi:10.3390/foods12040691_

Round 1

Reviewer 1 Report

The presented article is devoted to the technology of high pressure meat products on the example of rabbit myosin gels. This is a typical work for Food Science with the use of the entire set of experimental methods used to characterize food gels.

I think that the article can be improved taking into account the following comments and suggestions:

- In the review, the authors wrote: ”No reports have been reported on the use of konjac glucomannan and sodium caseinate to study high pressure treated rabbit myosin protein gels”. Here, it would be interesting to know if there were similar works, but with a different food protein, not with rabbit myosin. Much attention is paid mainly to konjac glucomannan, but the literature on the gelation of myosins of various origins is not presented. It is also necessary to indicate whether it is possible to obtain gels of myosin proteins without using high pressure.

- In my understanding, the pressures of 200 and 500 MPa are both high pressures. Therefore, the expression low pressure of 200 MPa sounds strange. In my opinion, it is better to use term mean pressure for the pressure of 200 MPa.

- Figures 1-7. For the indication of significantly different values, it is better use a1 instead of a1. The capital and lowercase letters are also often used in this case (Aa).

- Figure 4: Without error intervals, it makes no sense to quote symbols a,b,c. Figure 2: Symbol b2 or c2 is lost for the C+SC+mKGM.

Reviewer 2 Report

Review on manuscript: foods-2166947

 Effects of combining high pressure processing treatments and konjac glucomannan, sodium caseinate on gels properties of myosin protein

by  Yingying Cao, Lila Zhao and Huaiyu Li

submitted to Foods

In the manuscript submitted for comments, the authors studied the effect of combining high pressure processing treatments and konjac glucomannan, sodium caseinate on gels properties of myosin protein.

In my opinion, the purpose of the research and the connection between the results and practice are unclear. Also, I consider the scope of research to be limited and based only on the TPA test, and other methods could be used here, e.g. simple compression and stress-strain relationship, or a more advanced assessment of viscoelastic properties based on a creep or relaxation test. There is also no evaluation of the stability of such gels and their susceptibility to syneresis.

Detailed recommendation

Abstract – should be extended, what does better gel properties or best gel properties mean? more elasticity? better time or thermal stability? less syneresis?

keywords – missing myosin protein,

Introduction – abbreviation TG should be explained, for Latin name Italic style should be used, the abbreviation KGM should be used consistently, so not kgm.

The purpose of the work - it should be clearer, and should not be a summary of the gel preparation methodology, the authors should clearly indicate the purposefulness and practical aspect of such research, because it is an analysis of the textural properties of the hydrocolloids mixture.

TPA analysis – what was the shape and dimensions of the samples? force and hardness should be expressed in Newtons, adhesiveness should be expressed in N ∙mm. Gumminess applies only to semi-solid products and chewiness applies only to solid products.  Gumminess is mutually exclusive with chewiness since a product would not be both a semi-solid and a solid at the same time.

WBC – all abbreviations in the equation should be explained.

Results and discussion –  in my opinion, showing texture parameters in seven different figures is unjustified and should be replaced with a table, figures 4-6 – see notes earlier.

Conclusion – what does TPA properties or TPA property mean?

Reference 8 – is not clear.

Reference 23 – capital letter in journal title should be used.

Reviewer 3 Report

The work entitled “Effects of combining high pressure processing treatments and konjac glucomannan, sodium caseinate on gels properties of myosin protein” is interesting, however, this manuscript before acceptance should be improved (corrected), especially the discussion and methodological part. Materials and methods should be completed with more descriptions and details about methods and literature sources.  In addition, the superscripts of statistics in tables should be ordered.

Reviewer's suggestions below:

Regarding Keywordskeywords are partially repeating of the title, and it should be changed.

Regarding Introduction

All abbreviations should be explained e.g. "TG"

„No literatures have been reported about springiness, adhesiveness, cohesiveness, gumminess, chewiness, resilience, and ultra microstructure of mixing konjac glucomannan and sodium caseinate in rabbit myosin gels.” - this sentence should be deleted

”This research will make a contribution toward the improvement in the quality of high pressure meat products.” - this sentence should be deleted

Regarding Sample preparation

L: “……according to the methods described by Wang et al. and Hermansson et al. [23-24] with some modifications.” – Please describe, what kind of modifications have been done.

Table 1. - Unit should be added in the table. Below the table, all abbreviations, should be explained, e.g. mKGM

“My+SC:KGM(2:1)” or „My+SC:KGM(1:0.5)?

How many times gels were made?

2.3. Texture profile analysis (TPA)

“The deformation distance was 15 mm with trigger force 5 g and the gels were compressed twice.” – There are no literature, should be corrected.

Samples were compressed with P5 probe, but it is not clear, how the samples look like - its parameter`s (e.g. weight of samples, height in  glass beaker)

2.4. Water binding capacity (WBC)

Why, frozen samples were centrifuged - please explain.

“With regard to gels LP+H, centrifuging for 10min at 5000 g at 4 ºC, while gels HP, centrifuging for 10min at 10000g at 4 ºC.”  why different methodologies were used? Please explain.

"Measurements were carried out in triplicate on days 1 and 14 of refrigeration storage" what parameters were used?

Regarding Results and discussion

3.1. Texture profile analysis (TPA)

The authors examined, and described many parameters of texture, but only hardness was discussed with literature. In the chapter (except Hardness) there is no discussion of the results with the scientific literature

“The HP sample springiness values varied from 0.2 to 0.871 mm” replace with:  “The HP sample springiness values varied from 0.200 to 0.871 mm”

The gumminess properties of kinds of gels were showed in Fig 6. replace with: “The chewiness properties of kinds of gels were showed in Fig 6.”

”There was no pub-lished paper about resilience about rabbit myosin, SC and KGM of gel.” this sentence should be deleted

Notes to Figures and Tables:

The abbreviations in the table and under the figures should be explained.

Fig.2.: Sample „C+SC+mKGM” –„LP+H” – there are no statistical marks

Fig 5. – it is poor quality and it should be corrected.

Tables should be corrected, e.g. Table 1: „62.73±2.94a1” replace with: „62.73 a1 ±2.94”. Similarly Table 2.

“Table 2. WBC (water binding capacity) of myosin gels after 1 and 14 days of chilled storage.” replace with: “a-c means in a same column with different letters are significant different().” replace with: a-c means in a same column with different letters are significant different (P<0.05).”

Round 2

Reviewer 2 Report

After reassessing the manuscript Foods- 2166947v2 entitled "Effects of combining high pressure processing treatments and konjac glucomannan, sodium caseinate on gels properties of myosin protein" by Yingying Cao, Lila Zhao, Huaiyu Li, I can state that authors made necessary corrections taking under consideration mostly of my recommendations. But on a few points I do not share the opinion of the authors:

1.     the scope of performed analyzes is still limited,

2.     gumminess and chewiness should be used interchangeably, not together,

3.     lines 127-128 – sample preparation for TPA analysis is still unclear,

4.     the unit of force and hardness is Newton, and the unit of adhesiveness is N∙mm,

5.     replacing one figure with tables does not make sense.
